# Interpretable, non-mechanistic forecasting using empirical dynamic modeling and interactive visualization

**Lee Mason**[1]*, **Amy Berrington de Gonzalez**[2], **Montserrat Garcia-Closas**[2], **Stephen J. Chanock**[2], **Blànaid Hicks**[1], **Jonas S. Almeida**[2]

**1** Queen's University Belfast, Belfast, United Kingdom, **2** National Institutes of Health, Rockville, Maryland, United States of America

\* leekmason@gmail.com, masonlk@nih.gov

**Data Availability Statement:** The example data used in the tool is a combination of two CDC datasets: 'Weekly Counts of Deaths by State and Select Causes, 2014-2019', available at https://

## Abstract

Forecasting methods are notoriously difficult to interpret, particularly when the relationship between the data and the resulting forecasts is not obvious. Interpretability is an important property of a forecasting method because it allows the user to complement the forecasts with their own knowledge, a process which leads to more applicable results. In general, mechanistic methods are more interpretable than non-mechanistic methods, but they require explicit knowledge of the underlying dynamics. In this paper, we introduce EpiForecast, a tool which performs interpretable, non-mechanistic forecasts using interactive visualization and a simple, data-focused forecasting technique based on empirical dynamic modelling. EpiForecast's primary feature is a four-plot interactive dashboard which displays a variety of information to help the user understand how the forecasts are generated. In addition to point forecasts, the tool produces distributional forecasts using a kernel density estimation method–these are visualized using color gradients to produce a quick, intuitive visual summary of the estimated future. To ensure the work is FAIR and privacy is ensured, we have released the tool as an entirely in-browser web-application.

## Introduction

It is difficult to predict the future. Many forecasts fail to beat even basic benchmarks such as not projecting any changes at all [1], despite the fact that forecasters have access to more data and methods than ever before. The main issue is that the real-world is very complicated, it involves a lot of elements interacting in complex, non-linear ways [2–4]. Leading forecasting methods are often ineffective at dealing with this complexity, in part because they impose unrealistic or over-simplistic assumptions upon the data [5, 6]. On the other hand, there is limited evidence that complex methods lead to better forecasts than simple methods [7]. One way in which people attempt to make better forecasts is by using explicit, mechanistic models of the system dynamics which they wish to predict [8]. For example, in infectious epidemiology, an SIR (susceptible-infected-recovered) model aims to find the rates of change between three population groups: susceptible, infected, and recovered [9]. However, a mechanistic approach

data.cdc.gov/NCHS/Weekly-Counts-of-Deaths-by-State-and-Select-Causes/3yf8-kanr, and 'Weekly Provisional Counts of Deaths by State and Select Causes, 2020-2022', available at https://data.cdc.gov/NCHS/Weekly-Provisional-Counts-of-Deaths-by-State-and-S/muzy-jte6. The code is available at https://github.com/episphere/forecast under the MIT license. This includes the specific code for the EpiForecast website, reusable functions to perform EDM on the web, and JavaScript classes for the interactive plots. An Observable Notebook which shows how the code can be imported and used is available at https://observablehq.com/@siliconjazz/edm-interpretable-forecasting.

**Funding:** The author(s) received no specific funding for this work.

**Competing interests:** The authors have declared that no competing interests exist.

requires the forecaster to choose an appropriate model and correctly parameterize it, which is only possible if they have sufficient knowledge of the dynamics. Insufficient knowledge may lead the forecaster to choose an incorrect or over-simplistic mechanistic model, resulting in poor forecasts [10–13]. Indeed, even when the correct mechanistic model is chosen, it may be outperformed by a well-tuned model-free (non-mechanistic) method [14].

Non-mechanistic methods differ from mechanistic methods in that they do not require an explicit model of the dynamics [15, 16]. As such, they can be similarly applied to data from any domain so long as the data meets the method's assumptions. Non-mechanistic methods include statistical modeling (e.g. ARIMA, exponential smoothing) [17], empirical dynamic modeling (e.g. simplex) [18], and deep learning (e.g. LSTM) [19, 20]. Non-mechanistic methods are powerful and flexible, but they are generally less interpretable than mechanistic methods because their parameters lack a domain-specific meaning [15]. Interpretability is useful because it makes the relationship between the data and the forecasts more transparent. If a forecasting method is complex and difficult to interpret, it can give the user an inflated confidence in the quality of the forecasts that cannot be easily checked by the representation of the data [7]. When a complex method makes a surprising forecast, it is difficult for the user to assess whether that forecast is a realistic consequence of some hidden pattern in the data, or merely a fit to the noise. Similarly, if a forecasting method is interpretable with interactive visual reference to the data, it is easier for a forecaster to incorporate their knowledge into the forecasts (i.e. to "own the forecasts"), leading to more applicable results and thus more effective applications [21–23].

In this paper, we introduce EpiForcast: a web tool which produces detailed forecasts using an easy-to-understand, non-mechanistic approach. The tool uses interactive visualization to ensure interpretability, which allows the user to assess and improve the forecasts using their own knowledge. To ensure EpiForecast is accessible to a variety of users, we have released it as an open-source, in-browser web application. EpiForecast can work privately with local data or pull data from online data sources, allowing users to engage with live datasets. We have also made the tool available in a reactive notebook [24], complete with additional interactive explanations of the underlying principles.

## Results

### Overview

We have produced a web-tool which uses interactive visualization and empirical dynamic modeling to make interpretable, non-mechanistic forecasts. The tool is available at https://episphere.github.io/forecast and is also available, with additional explanations, in an Observable Notebook at https://observablehq.com/@siliconjazz/edm-interpretable-forecasting. We encourage readers to use these resources before reading the rest of this section because it will make the following explanations clearer. The tool is built on vanilla, client-side JavaScript using a small number of libraries, most notably D3. The tool consists of four interactive plots with some additional controls (see Fig 1). The basic principles of the tool are based on empirical dynamic modeling (EDM), a simple and intuitive non-mechanistic technique which is used for forecasting, especially of non-linear systems [18, 25]. The EDM process is relatively straightforward. It first searches for the nearest dynamic neighbors, moments in the past where the dynamics most closely resemble the recent dynamics. Then it then looks at what happened after each of these moments and uses that to predict what may happen next (see Methods for a detailed explanation). Each neighbor is weighted by how closely it resembles the most recent dynamics. This information can be used to generate point forecasts. The simplest way to accomplish this, used here, is by calculating weighted mean of all the neighbors' futures

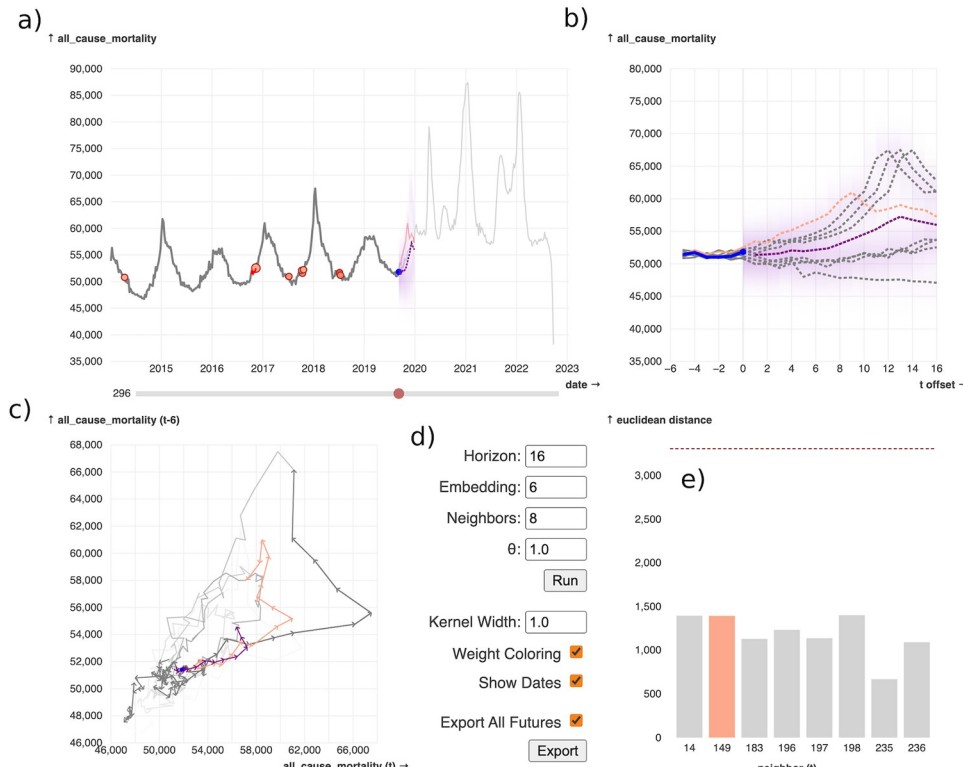

**Fig 1. Screenshot of the tool, available at https://episphere.github.io/forecast.** The example data depicted here is all-cause mortality from the Centers of Disease Control and Prevention (CDC). The user can also upload their own data or point the tool to data at a URL. **a)** The chronological time series, which shows the data accompanied by relevant information from the method. The blue dot is the current point, which is configurable by the user using the time-slider below the plot. The red dots are the nearest dynamic neighbors; their color is proportional to the respective neighbor's weight. The purple shading to the right of the current point is the forecast area, a continuous combination of the neighbors' futures. Darker areas in the forecast area are higher weighted, meaning they occur in more or higher weighted neighbors' futures. The purple dashed line is drawn by joining the point forecasts in the forecast horizon; the point forecasts are a weighted average of the neighbors' futures. **b)** An offset time-series plot which shows each of the neighbors and their futures, all offset to be on top of each other. This plot is useful for directly comparing the embedded neighbors and their futures. **c)** A phase-space plot, useful for inspecting dynamic patterns in the data (such as cycles). The phase space plot also clearly shows when the data has entered a new dynamic space, a case in which the EDM method will be less useful. **d)** Hyperparameters for the methods used by the tool, as well as a couple of additional visual options. **e)** A bar plot showing the distance of each neighbor to the current embedded point. This is useful for seeing explicitly which neighbors are closest, as well as seeing how that relates to their assigned color across the plots. The dashed line is the mean of all neighbors up to the selected current point in time. If a user hovers over one of the neighbors in any of the plots, then it is highlighted in all other plots. This allows the user to gain an immediate and concurrent sense of several aspects of the neighbor.

(see simplex method in Methods). However, in this tool we use visualization to deemphasize point predictions in favor of a more detailed representation of the range of potential futures which arise from the different dynamic neighbors.

## Details of the tool

The core of EpiForecast is an interactive dashboard comprised of four plots: a traditional time-series plot, an offset time-series plot, a phase space plot, and a bar plot (Fig 1). These plots are accompanied by additional input elements which control various aspects of the tool, including the hyperparameters of the forecasting method. The tool shows CDC mortality data by default, but the user can provide their own time-series data using the data configuration element.

Using the time slider beneath the traditional time-series plot, the user can select the time-step from which forecasts are made—we will refer to this as the "current" time-step. The main plot of the tool is a time-series plot marked with additional information (Fig 1A): the current time-step is marked with a purple dot, and the time-steps of the nearest neighbors are marked with red dots. The shade of red of each neighbor's dot is proportional to the weight of that neighbor; higher weighted neighbors are a darker shade. To the immediate right of the current time-step is a shaded forecast area which extends *tp* timesteps into the future (see Methods for more details about the forecasting method parameters). The purpose of this area is to summarize, at a glance, the offset futures of all nearest neighbors. In brief, the higher opacity parts of this area are closer to the offset futures of (higher-weighted) neighbors. More detail about how this area is generated can be found in *Methods*. Optionally, the user can choose to display point forecasts, which are represented as a dashed purple line. Point forecasts are another way to summarize the neighbors' futures, but they are less detailed than the visual representation provided by the shaded forecast area (see Fig 2).

The traditional time-series plot is accompanied by an offset time-series plot (Fig 1B) which shows the most recent dynamics (the embedded vector of the current time-point), the nearest neighbors, the neighbors' futures, the shaded forecast area, and (optionally) point forecasts. It is, essentially, a zoomed in version of the traditional time-series plot, with each neighbor offset such that the neighbors and their futures can be directly compared. This allows the user to get a better idea of how point forecasts and the shaded forecast area are produced from the nearest neighbors and their immediate futures. Another plot in the tool is the phase-space plot (Fig 1C), which can be used to inspect dynamic patterns such as the cyclic patterns present in the mortality data. This plot also helps highlight when the dynamics deviate from previously observed patterns—for example the COVID-19 pandemic substantially changed the dynamics of the mortality data, and consequently a new region of the phase space plot was populated. Finally, the tool contains a bar plot which shows the Euclidean distances between each dynamic neighbor and the recent dynamics (Fig 1E). To contextualize these values, a dashed line shows the mean distance between each previous time-step's recent dynamics and corresponding nearest neighbors.

Interaction is the critical feature of this tool. The central tenet argued in this report is that engaging multiple direct representations of the forecasts interactively leads to a deeper understanding of the underlying dynamics. To this end, EpiForecast uses the popular linking-and-brushing model in which a user can inspect a visual element in one plot and the highlight is reflected in the other plots [26]. For example, if the user hovers their mouse over a nearest neighbor dot on the traditional time-series plot, then the coloring of the visual elements in each of the other plots is changed to highlight that same neighbor. The same is true for the other plots. The traditional time-series plot also adheres to the "details on demand" principle of interactive visualization wherein additional details about a specific element are provided when the user requests them through interaction [27]. Specifically in EpiForecast, if the user hovers their mouse over a nearest neighbor dot in the traditional time-series plot then its future is displayed next to the current time-point, and a tooltip appears providing more details about the neighbor's distance and weight. The user can also right-click on a nearest neighbor (in any of the four plots) to disable it, meaning it will no longer by considered when generating the shaded forecast area or calculating the point forecasts. The time-slider beneath the traditional time-series plot (Fig 1A) is another useful way to interact with the tool because it lets the user explore forecasts at previous time-steps. This allows the user to both gain a better understanding of the forecasting method and to assess the suitability of the method for their data.

The plots are accompanied by a panel of additional controls (Fig 1D) which allow the user to control the hyperparameters of the EDM method, the relative width of the gaussian kernels

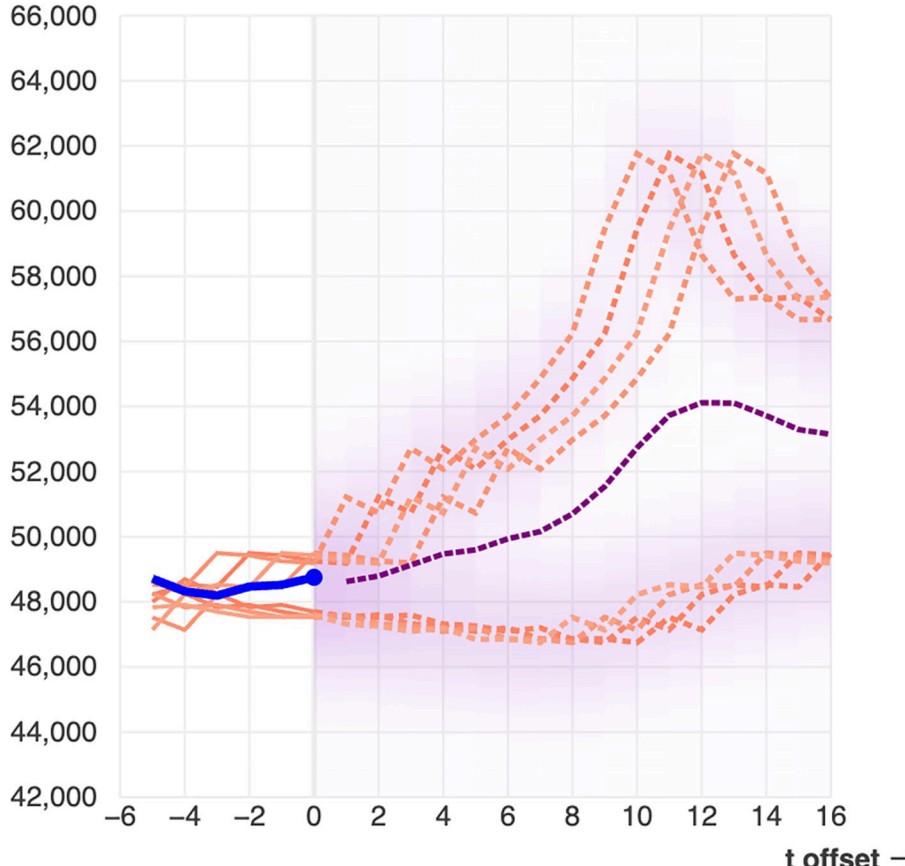

**Fig 2. This snapshot of the tool (using the default mortality data and with time point t = 87 selected) shows an example where point forecasts (the dashed purple line) are an ineffective summary of the forecasting results.** The forecasting method appears to identify two potential futures, one in which the number of deaths will rise to peak and then begin to decrease, and another in which the number of deaths decreases slowly before beginning to rise. In this case, point forecasts will fall between these two futures, meaning information about the two possibilities is lost in favor of a summary which is not representative of the method's more nuanced results. That nuance can be preserved by representing all neighbors' futures together, but that could become visually overwhelming with a larger number of neighbors or in more complex cases. We instead propose the shaded forecast area (the purple gradients in the forecast horizon) as a compromise. The shaded forecast area uses gaussian kernels to present a more detailed summary of the neighbors' futures than the point forecasts, while combining similar trajectories for the sake of visual simplicity.

used to generate the shaded forecast area, and some minor visual options. This panel also contains a button to export the results as either a series of point forecasts or as multiple parallel series, one for each neighbor. Instructions for interacting with the tool are available in a video linked on EpiForecast's web page and we encourage readers to play around with the interactive forecasts themselves to develop an understanding of how the interactivity facilitates better understanding.

It is difficult to compare the accuracy of EpiForecast with other forecasting techniques, because EpiForecast augments the forecasts with non-numeric concepts such as interaction and interpretability. However, it is useful to have a comparison with some basic benchmarks to justify certain design features, such as the shaded forecast area and the offset display of each neighbor's future. To accomplish this, we have produced a notebook at https://observablehq. com/@siliconjazz/epiforecast-dist-accuracy —the user can provide their own data to this

notebook. A copy of the results of the notebook on the default mortality data are available in S1 File.

## Example

Using Fig 1, we will illustrate how a detailed and deconstructed representation of forecasts can help an analyst generate applicable forecasts. In this example, it is 2019/09/06 and a public health official wishes to forecast US all-cause mortality rates (the example data for the tool). The forecaster notices that the method is drawing from several different situations in the past: the decreasing side of a peak in 2014, the increasing side of peaks in 2016 and 2017, and the bottom of a peak in 2018. The forecaster knows that unnormalized mortality counts tend to increase over time, and they therefore decide that the 2014 neighbor's contribution to the forecasts is not relevant, and thus they disable that neighbor. A number of influences in 2017/18 lead to a particularly deadly flu season (hence the larger peak in deaths). If the forecaster believed these influences were present again in 2019, they could disable the remaining neighbors in order to favor the neighbors from 2017. However, the forecaster is not sure, and thus leaves in the neighbors from 2016, 2017, and 2018. Finally, the forecaster exports the forecasts as 7 separate forecast series, one for each enabled neighbor. These exported series could now be analyzed in another environment using a suitable forecasting technique of the forecaster's choice, or they could form the basis of judgmental forecasts.

## Working privately with data

By default, the application shows the all-cause US mortality dataset from the Centers of Disease Control [28, 29], showcased in Fig 1, but users can supply their own data to the tool by uploading a CSV or JSON file, or by pointing the application to a corresponding URL. See the supplementary video or the tool's web page for more instructions on how to upload your own data to the tool. It is important to note that all visualizations and computations take place within the safety of the user's browser sandbox. No data or analytics circulate outside the user's Web browser, fully preserving privacy in order to enable the visualization of both public and sensitive data.

## Discussion

We have produced a FAIR web-tool which allows users to produce interpretable forecasts with their own data and then explore them in detail. This tool seeks to illustrate the role of interpretability via simplicity and interactivity, a key but often overlooked element of forecasting. If the forecaster can understand how a forecasting method produces results, then they can better assess the relevance and reliability of those results. As such, the primary goal of EpiForecast is not to produce more accurate point forecasts, but instead to further the "explainability" of forecasts by conveying a more detailed representation of the results—a representation which helps the user understand how the resulting forecasts were generated from the input data [21]. In a sense, this work treats forecasting as an exploratory process rather than an analytic one. Concrete numeric results are replaced by a more nuanced and detailed understanding of dynamic structures in the data and how these structures may be informative when forecasting. One way to achieve explainability is by choosing a method which in some way mirrors intuitive human reasoning. EDM is suitable for this because it based on the reasonable premise that immediate futures of similar pasts may provide insight into the future. EDM executes steps which are analogous to how a human may perform forecasting and, in doing so, produces a lot of information which is easy for a human to interpret. EpiForecast takes this information, visualizes it in multiple different ways, and makes it explorable through interaction.

EpiForecast uses the shaded forecasting area (see Fig 2) as a way to provide a more detailed representation of the information generated by EDM than would be gained from point forecasts. Interactivity is also used for this purpose. Point forecasts and the shaded forecast area are both ways to summarize the information from the EDM method, but with interactivity there is no need to summarize; we can just include all this information and the forecaster can view it on request. This provides the forecaster with a more complete understanding of the results and could thus reduce the bias which may arise from a static summary. The idea of using interaction visualizations to improve analytical insight is the core tenet of the field of visual analytics. There has been recent interest in using the principles of visual analytics to improve forecasting, and in particular to improve the forecaster's understanding of the forecasting method [21, 30].

Our accompanying tool has, nevertheless, several notable limitations. The EDM method requires a lot of data [31] and is most useful when the embedded space is well populated in the region of the current embedded point. Therefore, the tool will not work as well when the data has a substantial trend because the current embedded point will often fall in an uninhabited region of embedded space and therefore neighbors may not represent meaningfully similar pasts. As such, the tool will also be less effective for time-series where the long-term trend dominates the short-term dynamics, which usually occurs for long term time-series with few points. To some extent, this can be addressed by detrending the data, and the explainability of our tool makes it easier to identify other potentially useful preprocessing steps. Furthermore, the explainability allows a user to quickly see when the method is producing inappropriate forecasts, reducing the chance that they will be misled. EDM has several parameters which must be tuned, all of which can have a substantial effect on the visualization. However, these parameters are easy to interpret and can be quickly configured in the tool. This issue could potentially be addressed further by introducing an automatic parameter selection algorithm. A potential impedance to adoption of our tool is the fact that, paradoxically, complexity is often associated with depth and accuracy which leads users to more trust complex and difficult-to-interpret methods over simple and interpretable ones [7]. Finally, at present this tool works only on univariate data, but the EDM method can be extended to handle multivariate data so the tool could be updated to support this in the future.

We found a single example of another tool using interactive visualization and nearest neighbors for forecasting [32]. However, this tool does not find the neighbors in the embedded dynamic space, but instead in the multivariate space. The data has several variables which are relevant to forecasting energy demand, and the neighbors are single points in time which are closest to the current point in time along these variables. Like our tool, the energy tool uses visualization to highlight how, exactly, the neighbors are like the current time point. But because the tool does not consider dynamics the interactive plots differ substantially from our approach. However, it also suggests that approaching the multivariate dynamic space is a natural evolution of the work reported here.

To ensure that this tool is FAIR [33] and preserves the privacy of the data, we have developed the tool as an in-browser JavaScript application. Only the code needs to be hosted and computation is done on the client side, which means the application is easy and inexpensive to host. The web itself is a natural environment to host FAIR applications due to the ubiquity of the web-browser as both an interactive platform and an execution engine. Finally, due to the in-browser, client-side nature of the tool, the privacy of the user's data is ensured by default. In order to explore the participative modularity of this tool, its basic elements framed by interactive explanations are also made available in an Observable Notebook at https://observablehq.com/@siliconjazz/edm-interpretable-forecasting, which have important advantages over traditional notebooks [24, 34].

In conclusion, we have produced a tool which allows users to make forecasts using an non-mechanistic forecasting method and then explore those forecasts using interactive visualization. We hope that this tool will allow users to produce forecasts which are more informative, better understood, and more applicable to their respective domain. To ensure adherence to the FAIR principles, we have made this tool available as an open-source, entirely in-browser web application.

## Methods

### Empirical dynamic modeling (EDM)

Empirical dynamic modeling (EDM) is a time-series method which aims to empirically reconstruct the state-space of a system using delay embedding [18]. EDM can be used for a number of tasks, such as estimating the non-linearity of a system, but for our purposes we are interested in its use for forecasting. EDM has proven especially effective on complex, nonlinear systems, such as ecological models, but it requires a lot of data. EDM is effective at modeling univariate and multivariate data [18], but in this paper we will explore the univariate implementation. The input data is a univariate time-series with evenly spaced timepoints:

$$\overrightarrow{y} = [y_1, y_2, \ldots, y_T] \tag{1}$$

The first step is to perform the delay embedding: placing each value in a vector with the *E—1* values which precede it in time:

$$\overrightarrow{v_t} = [y_{t-E}, \ldots, y_{t-1}, y_t]. \tag{2}$$

Where $E$ (a hyperparameter) is the embedding dimension (the number of values in each embedded vector). Next, the algorithm finds the $n$ nearest neighbors of the most recent embedded vector $\overrightarrow{v_T}$, where $n$ is another hyperparameter. That is, the $n$ embedded vectors which have the least euclidean distance to $\overrightarrow{v_T}$,. Here we define $t_i$ to be the time-step of the $i^{\text{th}}$ neighbor, and $x_i$ to be the embedded vector of the $i^{\text{th}}$ neighbor $\overrightarrow{x_i} = \overrightarrow{v_{t_i}}$. Next, the algorithm retrieves the 'future vector' for each neighbor, a vector of values which immediately succeed the neighbor:

$$\overrightarrow{z}_i = [y_{t_i+1}, \ldots, y_{t_i+tp}]. \tag{3}$$

Where $tp$ is the forecast horizon, another hyperparameter which indicates how far into the future the algorithm will forecast. The algorithm then calculates a weight for each neighbor:

$$w_i = e^{-\frac{\theta\, d_i}{\bar{d}}} \tag{4}$$

where $d_i$ is the euclidean distance between $\overrightarrow{v_T}$ and the neighbor $\overrightarrow{x_i}$, and $\bar{d}$ is the mean euclidean distance between $\overrightarrow{v_T}$ and each neighbor. The variable θ is another hyperparameter which specifies the extent to which the weight is affected by the distance, if θ = 0 then all neighbors are weighted equally at 1. This covers the information used by the tool to draw the shaded forecast area, details of how this is achieved are provided in Section 4.3. However, the information can also be used to generate a vector of point forecasts: forecasts with a single, numerical value. There are a few ways to accomplish this, but we have chosen the simplex method due to its simplicity (18). The simplex method calculates a forecast vector by taking a weighted average of the neighbors' future vectors:

$$\overrightarrow{y_{T+tp}} = \sum_{i=1}^{n} w_i \cdot \overrightarrow{z_i} \tag{5}$$

## Drawing the shaded forecast area

The shaded forecasting area is generated using a method based on weighted kernel density estimation. At t = T we have n neighbors, each with a future vector $\vec{z}_i$ and a weight $w_i$. The following process is repeated for each timestep r in the forecast horizon: $T+1 \leq r \leq T+tp$. At $t = T + r$, the algorithm takes the corresponding value $z_{i,T+r}$ from each neighbor's future vector. At each value, the algorithm centers a gaussian kernel with height proportional to the neighbor's weight and width controlled by a hyperparameter $\alpha$. Specifically, at $t = T+r$ for the i$^{th}$ neighbor we have a kernel function:

$$K_{i,r}(v) = s \cdot w_i \cdot f\left(\frac{1}{\alpha} \cdot q \cdot (v - z_{i,T+r})\right) \tag{6}$$

Where $f$ is the probability density function of the normal distribution with mean 0 and variance 1, $\alpha$ is the width of the kernel, and $s$ and $q$ are scaling parameters for the height and width respectively. The scaling parameter s = $1/(f(0) \cdot n)$ is set to ensure the sum of all kernel peaks would be equal to 1 if all neighbors had a weight equal to 1. The "width" $\alpha$ is the width of the density function $f$ at probability density $c$–we set $c = 0.0001$, an arbitrary low value. The scaling parameter q is set $q = \sqrt{-\ln(2c^2\pi)}$ to ensure this width definition. The user can scale the width of the kernel relative to the default width using the hyperparameter kw: $\alpha = \sigma \cdot kw$. The overall equation for the shaded area is the sum over each kernel:

$$K_r(v) = \sum_{i=1}^{n} K_{i,r}(v) \tag{7}$$

Values are calculated for a range which covers all of the kernels, and each value is linearly mapped to a color on a color scale. These colors are used to draw a gradient at $t = r$ on the full and offset time-series plots. For a visual example of this process, see Fig 3.

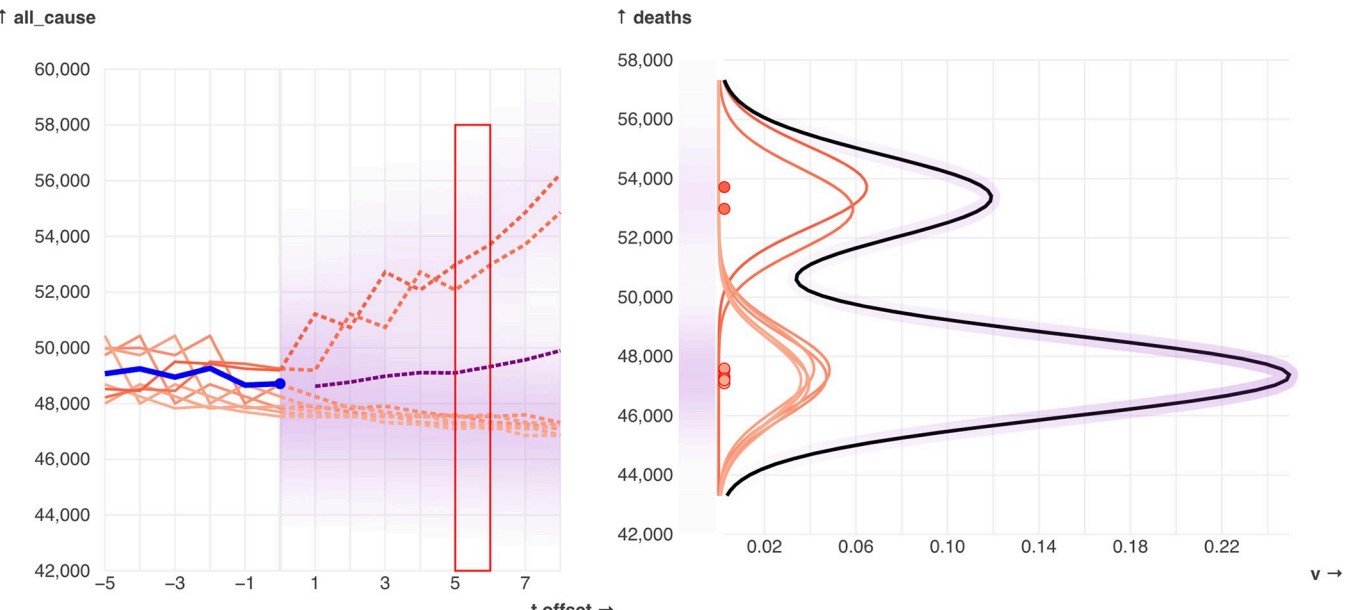

**Fig 3. An example of how the weighted kernel density estimation works, and how the resulting values are mapped onto colors for the shaded forecasting area.** The example is from t = 82 and tp = 8 on the default mortality data, with the relative kernel width parameter, kernel_width set to 1.0. A gaussian kernel is placed at each neighbor's value; higher weighted neighbors (represented with a darker color) correspond to taller and thinner kernels. The sum of the kernels forms the forecast distribution (black line) which is then linearly mapped onto an opacity value used to draw the color gradient.

## Supporting information

**S1 File. Distributional accuracy benchmarks.** Comparison of the elements of the method with some basic benchmarks.
(PDF)

## Author Contributions

**Conceptualization:** Lee Mason, Amy Berrington de Gonzalez, Montserrat Garcia-Closas, Stephen J. Chanock, Blànaid Hicks, Jonas S. Almeida.

**Data curation:** Lee Mason, Jonas S. Almeida.

**Formal analysis:** Lee Mason.

**Investigation:** Lee Mason, Jonas S. Almeida.

**Methodology:** Lee Mason, Amy Berrington de Gonzalez, Montserrat Garcia-Closas, Blànaid Hicks, Jonas S. Almeida.

**Resources:** Lee Mason, Amy Berrington de Gonzalez, Montserrat Garcia-Closas, Jonas S. Almeida.

**Software:** Lee Mason, Jonas S. Almeida.

**Supervision:** Amy Berrington de Gonzalez, Montserrat Garcia-Closas, Blànaid Hicks, Jonas S. Almeida.

**Validation:** Lee Mason.

**Visualization:** Lee Mason, Jonas S. Almeida.

**Writing – original draft:** Lee Mason.

**Writing – review & editing:** Lee Mason, Amy Berrington de Gonzalez, Montserrat Garcia-Closas, Blànaid Hicks, Jonas S. Almeida.

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
