## [Decision Letter · Decision Letter 0]

23 Jan 2023

PONE-D-22-28854Interpretable, non-mechanistic forecasting using empirical dynamic modeling and interactive visualizationPLOS ONE

Dear Dr. Mason,

Thank you for submitting your manuscript to PLOS ONE. After careful consideration, we feel that it has merit but does not fully meet PLOS ONE’s publication criteria as it currently stands. Therefore, we invite you to submit a revised version of the manuscript that addresses the points raised during the review process.

We look forward to receiving your revised manuscript.

Kind regards,

Zakariya Yahya Algamal, PhD

Academic Editor

PLOS ONE

Journal Requirements:

Reviewers' comments:

Reviewer's Responses to Questions

**Comments to the Author**

1. Is the manuscript technically sound, and do the data support the conclusions?

Reviewer #1: Partly

Reviewer #2: Yes

2. Has the statistical analysis been performed appropriately and rigorously? 

Reviewer #1: Yes

Reviewer #2: No

3. Have the authors made all data underlying the findings in their manuscript fully available?

Reviewer #1: Yes

Reviewer #2: Yes

4. Is the manuscript presented in an intelligible fashion and written in standard English?

Reviewer #1: Yes

Reviewer #2: Yes

5. Review Comments to the Author

Reviewer #1: 1. Abstract should be rewritten correctly with more details about proposed and used methods.

2. Methods section doesn't have sufficient details and basic equations about proposed methods.

3. In results section, statistical error measurements should be used for comparing the accuracy of result and inserted into tables.

4. Conclusions should be rewritten in more appropriate contents.

Reviewer #2: The authors provided a web-application as a tool to produce the interpretations of non-mechanistic forecasts using Empirical Dynamic Modelling and interactive visualization. This tool is based on empirical dynamic modelling (EDM), a simple and intuitive non-mechanistic technique with four interactive plots. They made this tool available online for researchers and more details. Results, discussions and the statistical method used have been introduced. The manuscript is technical without theoretical contribution. Novelty resides in providing a web-application.

In my opinion, the manuscript is interesting idea and scientifically coherent. It tackles an important issue. The state of the literature in the background is comprehensive and the related work is adequately cited. However, I find :

Difficulty in following all the information in the manuscript due to the presentation is not very clear to me.

The proposed tool should represent by an algorithm. It is not easy to understand the methodology of the tool through textual narration.

I encourage the authors to spend more time in the section on Methods to present the details through a mathematical framework, as well as, for the mathematical notations.

The authors full name in the reference should be included. For example, in reference (11) there are missing authors' names.

Finally, the details of interactive visualization are not presented as well.

6. PLOS authors have the option to publish the peer review history of their article (what does this mean?). If published, this will include your full peer review and any attached files.

Reviewer #1: No

Reviewer #2: No

---

## [Author Response · Author response to Decision Letter 0]

23 Feb 2023

Dear Dr. Algamal,

Thank you for inviting us to re-submit our work to PLOS ONE. We would also like to thank you and the reviewers for taking the time to consider our manuscript and provide helpful feedback. We are very grateful for the insightful comments raised by the reviewers, and we hope that the revised manuscript is suitable for publication in your journal. Below, we have listed the suggestions and the changes we made in response:

• “Difficulty in following all the information in the manuscript due to the presentation is not very clear to me.” (Reviewer 2)

We have updated several parts of the manuscript to improve clarity, especially the methods and results sections. 

• “The proposed tool should represent by an algorithm. It is not easy to understand the methodology of the tool through textual narration.” (Reviewer 2)

• “I encourage the authors to spend more time in the section on Methods to present the details through a mathematical framework, as well as, for the mathematical notations.” (Reviewer 2)

• “Methods section doesn't have sufficient details and basic equations about proposed methods.” (Reviewer 1)

In the methods section, we have moved many of the inline equations to their own lines, and updated the surrounding narration to clarify the role each equation plays in the tool. 

• “Finally, the details of interactive visualization are not presented as well.” (Reviewer 2)

A very helpful comment, and we agree completely that a specific account of the interactive plots was missing in the original manuscript. To remedy this, we have re-written and expanded much of the results section, specifically to include more details of the interaction.

• “The authors full name in the reference should be included. For example, in reference (11) there are missing authors' names.” (Reviewer 2)

The references in our report are in the Vancouver style, as suggested by the PLOS ONE guidelines. The Vancouver style has a unique way of representing authors name where their last name is followed by their initials, without periods between the initials (e.g. for me it would be Mason LK). We (myself and another co-author) were unfamiliar with this style prior to submission but we have re-checked the references and we believe they adhere to this style. Regarding reference (11) in particular, the name “Pe’er G” perhaps appears abbreviated, but that is the spelling listed on the paper. 

• “Abstract should be rewritten correctly with more details about proposed and used methods.” (Reviewer 1)

We have re-written the abstract to include a specific reference to the methods used (empirical dynamic modelling) and the methods proposed (the interactive dashboard and the shaded forecasting area based on kernel density estimation).

• “In results section, statistical error measurements should be used for comparing the accuracy of result and inserted into tables.” (Reviewer 1)

In the results section, we have now included a link to a notebook which compares the accuracy of the kernel density estimation method to some basic benchmarks. We have also included a tabular form of these comparisons (for the default data) in Supporting Information S1; this is also referenced in the results section. 

• Conclusions should be rewritten in more appropriate contents. (Reviewer 1)

We have rewritten the conclusion paragraph so that it relates better to the rest of the manuscript. 

Once again, we’d like to thank the reviewers for these helpful comments. We are delighted to have to resubmit this paper with the suggested changes. We look forward to hearing your response.

Sincerely,

Lee Mason 

Division of Cancer Epidemiology and Genetics, National Cancer Institute 9609 Medical Center Drive, Rockville, Maryland 20850

Email: masonlk@nih.gov

---

## [Decision Letter · Decision Letter 1]

16 Mar 2023

Interpretable, non-mechanistic forecasting using empirical dynamic modeling and interactive visualization

PONE-D-22-28854R1

Dear Dr. Mason,

We’re pleased to inform you that your manuscript has been judged scientifically suitable for publication and will be formally accepted for publication once it meets all outstanding technical requirements.

Kind regards,

Zakariya Yahya Algamal, PhD

Academic Editor

PLOS ONE

Additional Editor Comments (optional):

The authors have adequately addressed all the comments raised in a previous round of review.

Reviewers' comments:

Reviewer's Responses to Questions

**Comments to the Author**

1. If the authors have adequately addressed your comments raised in a previous round of review and you feel that this manuscript is now acceptable for publication, you may indicate that here to bypass the “Comments to the Author” section, enter your conflict of interest statement in the “Confidential to Editor” section, and submit your "Accept" recommendation.

Reviewer #2: All comments have been addressed

2. Is the manuscript technically sound, and do the data support the conclusions?

Reviewer #2: Yes

3. Has the statistical analysis been performed appropriately and rigorously? 

Reviewer #2: Yes

4. Have the authors made all data underlying the findings in their manuscript fully available?

Reviewer #2: Yes

5. Is the manuscript presented in an intelligible fashion and written in standard English?

Reviewer #2: Yes

6. Review Comments to the Author

Reviewer #2: The methods used in the study are appropriate and well-described. The discussion section provides an in-depth interpretation of the results.

7. PLOS authors have the option to publish the peer review history of their article (what does this mean?). If published, this will include your full peer review and any attached files.

Reviewer #2: No

---

## [Editor Report · Acceptance letter]

23 Mar 2023

PONE-D-22-28854R1 

Interpretable, non-mechanistic forecasting using empirical dynamic modeling and interactive visualization 

Dear Dr. Mason:

I'm pleased to inform you that your manuscript has been deemed suitable for publication in PLOS ONE. Congratulations! Your manuscript is now with our production department. 

Kind regards, 

on behalf of

Professor Zakariya Yahya Algamal 

Academic Editor

PLOS ONE